# Relational Causal Models with Cycles: Representation and Reasoning

**Ragib Ahsan** RAHSAN3@UIC.EDU

**David Arbour** ARBOUR@ADOBE.COM

**Elena Zheleva** EZHELEVA@UIC.EDU

**Editors:** Bernhard Schölkopf, Caroline Uhler and Kun Zhang

## Abstract

Causal reasoning in relational domains is fundamental to studying real-world social phenomena in which individual units can influence each other's traits and behavior. Dynamics between interconnected units can be represented as an instantiation of a relational causal model; however, causal reasoning over such instantiation requires additional templating assumptions that capture feedback loops of influence. Previous research has developed lifted representations to address the relational nature of such dynamics but has strictly required that the representation has no cycles. To facilitate cycles in relational representation and learning, we introduce relational $\sigma$-separation, a new criterion for understanding relational systems with feedback loops. We also introduce a new lifted representation, $\sigma$-*abstract ground graph* which helps with abstracting statistical independence relations in all possible instantiations of the cyclic relational model. We show the necessary and sufficient conditions for the completeness of $\sigma$-AGG and that relational $\sigma$-separation is sound and complete in the presence of one or more cycles with arbitrary length. To the best of our knowledge, this is the first work on representation of and reasoning with cyclic relational causal models.

**Keywords:** Representation, Cycles, Relational Causal Model

## 1. Introduction

Causal inference methods from observational data often assume that causal relationships can be represented in an acyclic graphical model. However, many real-world phenomena involve feedback loops or cycles that violate the acyclicity assumption. For example, supply and demand affect price and vice versa, hormone levels in the body affect each other and friends can impact each other's choices. Common to these scenarios is that these interactions occur over time but any individual directed relationship may not be observed. Instead what is observed is a set of values that are result of long term individual interactions. This current state can be represented through a cycle.

The existing works on cyclic causal models primarily focus independently and identically distributed (i.i.d) data instances (Richardson, 1996, 1997; Strobl, 2019; Rantanen et al., 2020). However, in many real-world systems units are often interconnected in a complex network. Causal reasoning over such relational systems is central to understanding real-world social phenomena, such as social influence and information diffusion. For example, a well-known study by Christakis and Fowler (2007) investigates whether obesity is contagious in a population where each person's eating habits can affect their friends and family and vice-versa. Similarly, several real-world problems in epidemiology and computational social science encounter such mutual influence and interaction

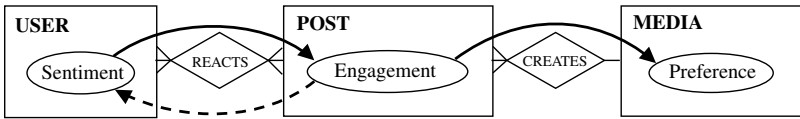

Figure 1: Example of relational model with and without feedback loop. Rectangle, rhombus and oval shapes represent entity, relationship and attributes respectively. Arrows refer to relational dependence. The solid arrows constitute an acyclic relational model. The dashed arrow creates a feedback loop in the model.

among human subjects. For example, effect of vaccination in mother and child (Shpitser, 2015; VanderWeele et al., 2012), and justices influencing each other in supreme court decisions (Ogburn et al., 2020) are all examples of mutual influence. Studying such phenomena requires causal reasoning over complex interactions between interconnected entities. However, causal questions of mutual influence in a relational system can be hard to answer due to the lack of appropriate causal representations and reasoning mechanisms.

One way to represent interference is through SCMs which capture pairwise relationships (Ogburn et al., 2014; Shalizi and Thomas, 2011), where one individual's influence on another is represented independently. Such representation is easy to reason about but it assumes that pairs of nodes are independent of other pairs of nodes. Unfortunately, this does not reflect the properties of real-world social networks in which nodes are arbitrarily interconnected with each other.

The development of *relational causal models*, which generalize over structural causal models, is an important step towards capturing interactions between non-i.i.d instances (Maier et al., 2013b,a; Lee and Honavar, 2015; Bhattacharya et al., 2020). Relational models involve multiple types of interacting entities with probabilistic dependencies among their attributes. Maier et al. (2013b) developed a lifted causal representation named *abstract ground graph (AGG)* that abstracts over all instantiations of a relational model. AGG enables reasoning about causal queries in relational causal models and relational causal discovery. However, existing relational causal models assume acyclicity and do not allow for reasoning about identification in the presence of feedback loops.

Figure 1 shows an example of a relational model where users react to news articles or posts on a specific topic (e.g., vaccines) generated by media agencies. The preference of the media regarding a topic (e.g., pro- vs anti-vaccination) is influenced by the engagement or feedback (e.g., positive/negative comments) it receives on its existing posts. The sentiment of a user towards a given post directly impacts their engagement in the post. The relational model representation tempts one to conclude that the sentiment of users regarding vaccination is independent of the preference of media agencies given engagements of the posts the users react to. However, as we show in section 3.5, this is not necessarily true. Moreover, users' sentiment can also be impacted by the engagements in posts they interact with. The dashed arrow in Figure 1 represents such a dependency which makes the model a cyclic one. Unfortunately, even though the model seems simple and realistic, the abstract ground graph (AGG) representation and relational $d$-separation no longer apply due to the presence of a feedback loop (i.e., a cycle).

In this work, we specifically study cyclic RCMs and show that they offer the necessary representation to reason about a lot of real-world causal problems where popular assumptions do not hold. We do not assume the Stable Unit Treatment Value Assumption (SUTVA), of causal inference, according to which the outcome of each unit depends only on the unit itself (Rubin, 1980).

This Assumption is violated in relational systems where the treatment and outcome of one unit can impact other units' outcomes. To the best of our knowledge, this is the first work that addresses representation of and reasoning about cyclic relational causal models. We define a new abstract representation, *$\sigma$-abstract ground graph* ($\sigma$-AGG) which generalizes over cyclic relational models. In order to reason about relational queries in $\sigma$-AGG, we introduce *relational $\sigma$-separation* and provide proofs for its soundness and completeness for all instantiations of a relational model. The implications of this new representation are twofold. First, it can lead the way for reasoning about causal effects like interference and contagion in relational systems, which are of wide interest in the social sciences. Second, the abstract representation and its identified properties will lay a foundation for causal structure learning from relational models with cycles. We show the sufficient conditions for the completeness of $\sigma$-AGG by first resolving an open problem on the completeness of AGG. Finally, we discuss the Markov condition of relational $\sigma$-separation and its implications.

## 2. Related Work

Maier et al. (2013b) proposed a sound and complete abstract representation for relational data with multiple entities and relationships. They defined an abstract representation of relational dependence called abstract ground graph based on the assumption that the underlying relational model is acyclic. Moreover, they introduced relational $d$-separation, a new criterion to reason about statistical independence in all the instantiations of a relational model. They proved the soundness and completeness of relational $d$-separation. The AGG and associated relational $d$-separation criterion together allows a formal basis of causal analysis in relational systems. The same authors proposed a constraint-based causal structural learning algorithm called RCD (Maier et al., 2013a) which is first of its kind.

Lee and Honavar (2015) presented a critical view on the generalizability of AGG where they constructed a counterexample which shows how AGG can fail to abstract all $d$-separation relationships in the ground graphs. They also identified a shortcoming of the necessary conditions for considering a intersection variable and provided a revised set of conditions. In a different work, Lee and Honavar (2020) proposed a new abstract representation of relational causal models called the unrolled graph which guarantees a weaker sense of completeness. However, neither of the two groups of work moved away from the assumption of acyclicity. As a result their developments are not directly applicable to relational models with cycles or feedback loops. Alternatively, *chain graphs* have been used for representing systems with pairwise feedback loops (Lauritzen and Richardson, 2002; Dawid, 2010; Ogburn et al., 2020). There is no prior work that focuses on representation and reasoning of relational models with cycles or feedback loops of arbitrary length.

Existing research on representation and reasoning about systems with cycles is based on propositional variables, not relational ones. Spirtes (1995) showed that, in general case, without any specific assumption regarding the nature of dependence (linear, polynomial etc.), the $d$-separation relations in a directed cyclic graph are not sufficient to entail all the corresponding conditional independence relations. Forré and Mooij (2017) introduced $\sigma$-separation, an alternative formulation for which the corresponding Markov property is shown to hold in a very general setting. According to them, the $\sigma$-separation Markov property seems appropriate for a wide class of cyclic structural causal models with non-linear functional relationships between non-discrete variables.

## 3. Preliminaries

In this section we introduce the necessary terminology to discuss representation and abstraction of cyclic relational models.

### 3.1. Directed Cyclic Graphs

A Directed Cyclic Graph (DCG) is a graph $\mathcal{G} = \langle \mathcal{V}, \mathcal{E} \rangle$ with nodes $\mathcal{V}$ and edges $\mathcal{E} \subseteq \{(u, v) : u, v \in V, u \neq v\}$ where $(u, v)$ is an ordered pair of nodes. We will denote a directed edge $(u, v) \in \mathcal{E}$ as $u \rightarrow v$ or $v \leftarrow u$, and call $u$ a parent of $v$. In this work, we restrict ourselves to DCG as the causal graphical model. A walk between two nodes $u, v \in \mathcal{V}$ is a tuple $\langle v_0, e_1, v_1, e_2, v_2, ..., e_n, v_n \rangle$ of alternating nodes and edges in $\mathcal{G}(n \geq 0)$, such that $v_0, ..., v_n \in \mathcal{V}$, and $e_1, ..., e_n \in \mathcal{E}$, starting with node $v_0 = u$ and ending with node $v_n = v$ where the edge $e_k$ connects the two nodes $v_{k-1}$ and $v_k \in \mathcal{G}$ for all $k = 1, ..., n$. If the walk contains each node at most once, it is called a *path*. A *directed walk (path)* from $v_i \in \mathcal{V}$ to $v_j \in \mathcal{V}$ is a walk (path) between $v_i$ and $v_j$ such that every edge $e_k$ on the walk (path) is of the form $v_{k-1} \rightarrow v_k$, i.e., every edge is directed and points away from $v_i$.

We get the *ancestors* of node $v_j$ by repeatedly following the path(s) through the parents: $AN_{\mathcal{G}}(v_j) := \{v_i \in V : v_i = v_0 \rightarrow v_1 \rightarrow ... \rightarrow v_n = v_j \in \mathcal{G}\}$. Similarly, we define the *descendants* of $v_i : DE_{\mathcal{G}}(v_i) := v_j \in \mathcal{V} : v_i = v_0 \rightarrow v_1 \rightarrow ... \rightarrow v_n = v_j \in \mathcal{G}$. Each node is an ancestor and descendant of itself. A directed cycle is a directed path from $v_i$ to $v_j$ such that in addition, $v_j \rightarrow v_i \in \mathcal{E}$. All nodes on directed cycles passing through $v_i \in \mathcal{V}$ together form the strongly connected component $SC_{\mathcal{G}}(v_i) := AN_{\mathcal{G}}(v_i) \cap DE_{\mathcal{G}}(v_i)$ of $v_i$.

### 3.2. $\sigma$-separation

The idea of $\sigma$-separation follows from $d$-separation, a fundamental notion in DAGs which was first introduced by Pearl (1988):

**Definition 1 ($d$-separation)** *A walk $\langle v_0...v_n \rangle$ in DCG $G = \langle \mathcal{V}, \mathcal{E} \rangle$ is $d$-separated by $C \subseteq V$ if:*

1. *its first node $v_0 \in C$ or its last node $v_n \in C$, or*

2. *it contains a collider $v_k \notin AN_{\mathcal{G}}(C)$, or*

3. *it contains a non-collider $v_k \in C$.*

*If all paths in $\mathcal{G}$ between any node in set $A \subseteq \mathcal{V}$ and any node in set $B \subseteq \mathcal{V}$ are $d$-blocked by a set $C \subseteq \mathcal{V}$, we say that $A$ is $d$-separated from $B$ by $C$, and we write $A \underset{\mathcal{G}}{\overset{d}{\perp\!\!\!\perp}} B | C$.*

$d$-separation exhibits the global Markov property in DAGs which states that if two variables $X$ and $Y$ are $d$-separated given another variable $Z$ in a DAG representation then $X$ and $Y$ are conditionally independent given $Z$ in the corresponding distribution of the variables. However, Spirtes (1995); Neal (2000) show that without any specific assumption regarding the nature of dependence (i.e. linear, polynomial), the $d$-separation relations are not sufficient to entail all the corresponding conditional independence relations in a DCG. In a recent work, an alternative formulation called $\sigma$-separation is introduced which holds for a very general graphical settings (Forré and Mooij, 2017).

Here, we consider a simplified version of the formal definition of $\sigma$-separation:

**Definition 2 ($\sigma$-separation)** *(Forré and Mooij, 2017)*
*    A walk $\langle v_0...v_n \rangle$ in DCG $G = \langle \mathcal{V}, \mathcal{E} \rangle$ is $\sigma$-blocked by $C \subseteq V$ if:*

1. *its first node $v_0 \in C$ or its last node $v_n \in C$, or*

2. *it contains a collider $v_k \notin AN_{\mathcal{G}}(C)$, or*

3. *it contains a non-collider $v_k \in C$ that points to a node on the walk in another strongly connected component (i.e., $v_{k-1} \to v_k \to v_{k+1}$ with $v_{k+1} \notin SC_{\mathcal{G}}(v_k)$, $v_{k-1} \leftarrow v_k \leftarrow v_{k+1}$ with $v_{k-1} \notin SC_{\mathcal{G}}(v_k)$ or $v_{k-1} \leftarrow v_k \to v_{k+1}$ with $v_{k-1} \notin SC_{\mathcal{G}}(v_k)$ or $v_{k+1} \notin SC_{\mathcal{G}}(v_k)$).*

*If all paths in $\mathcal{G}$ between any node in set $A \subseteq \mathcal{V}$ and any node in set $B \subseteq \mathcal{V}$ are $\sigma$-blocked by a set $C \subseteq \mathcal{V}$, we say that $A$ is $\sigma$-separated from $B$ by $C$, and we write $A \overset{\sigma}{\underset{\mathcal{G}}{\perp\!\!\!\perp}} B | C$.*

Forré and Mooij (2017) show that the global directed Markov property [1] holds for $\sigma$-separation in directed cyclic graphs, unlike $d$-separation. This enables $\sigma$-separation to perform a similar role to $d$-separation but for directed cyclic graphs. $\sigma$-separation is a generalization of $d$-separation.

### 3.3. $\sigma$-faithfulness

$\sigma$-faithfulness refers to the property which states that all statistical dependencies found in the distribution generated by a given causal structure model is entailed by the $\sigma$-separation relationships.

**Definition 3 ($\sigma$-faithfulness)** *Given $\mathcal{X}_A$, $\mathcal{X}_B$, $\mathcal{X}_C$ as the distributions of variables A, B, C respectively in solution $\mathcal{X}$ of a causal model $\mathcal{M}$, $\sigma$-faithfulness states that if $\mathcal{X}_A$ and $\mathcal{X}_B$ are conditionally independent given $\mathcal{X}_C$, then A and B are $\sigma$-separated by C in the corresponding possibly cyclic graphical model $\mathcal{G}$ of $\mathcal{M}$.*

### 3.4. Relational Causal Models (RCM)

We adopt the definition of relational causal model used by previous work on relational causal discovery (Maier et al., 2013a; Lee and Honavar, 2020). We denote random variables and their realizations with uppercase and lowercase letters respectively, and bold to denote sets. We use a simplified Entity-Relationship model to describe relational data following previous work (Heckerman et al., 2007). A relational schema $\mathcal{S} = \langle \mathcal{E}, \mathcal{R}, \mathcal{A}, card \rangle$ represents a relational domain where $\mathcal{E}, \mathcal{R}$ and $\mathcal{A}$ refer to the set of entity, relationship and attribute classes respectively. It includes a cardinality function that constrains the number of times an entity instance can participate in a relationship. Figure 1 shows an example relational model that describes a simplified user-media engagement system. The model consists of three entity classes (User, Post, and Media), and two relationship classes (Reacts and Creates). Each entity class has a single attribute. The cardinality constraints are shown with crow's feet notation— a user can react to multiple posts, multiple users can react to a post, a post can be created by only a single media entity.

A *relational skeleton s* is an instantiation of a relational schema $\mathcal{S}$, represented by an undirected graph of entities and relationships. Figure 2a shows an example skeleton of the relational model from Figure 1. It shows that Alice and Bob both react to post P1. Alice also reacts to post P2. P1

---

1. Definition given in appendix

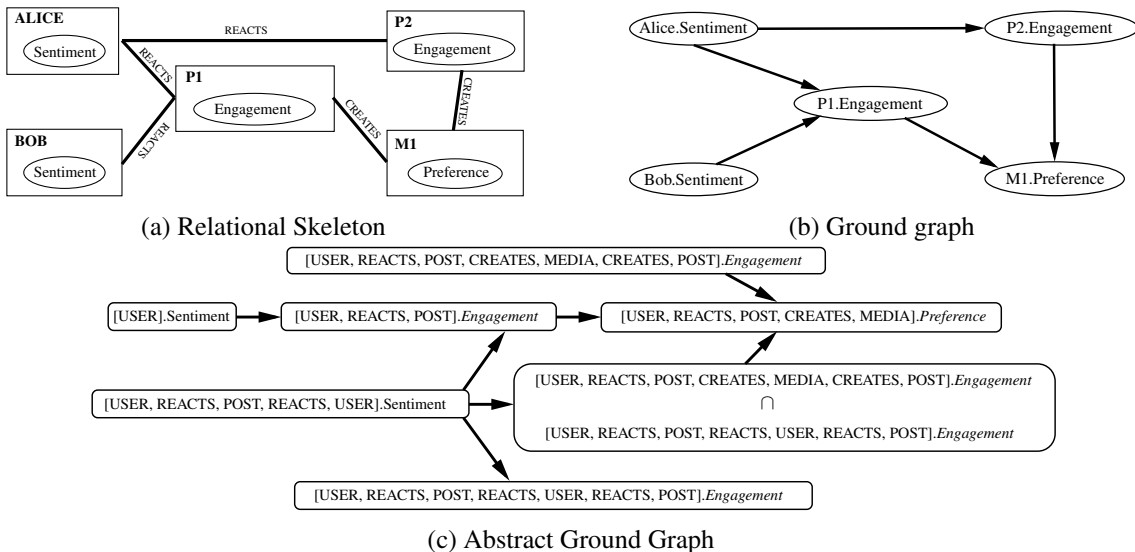

Figure 2: Fragments of a relational skeleton, ground graph, and abstract ground graph corresponding to the relational causal model from Figure 1. The arrows represent relational dependencies.

and P2 both are created by media M1. There could be infinitely many possible skeletons for a given RCM. We denote the set of all skeletons for schema $\mathcal{S}$ as $\sum_{\mathcal{S}}$.

Given a relational schema, we can specify relational paths, which intuitively correspond to ways of traversing the schema. For the schema shown in Figure 1, possible paths include $[User, Reacts, Post]$ (the posts a user reacts to), as well as $[User, Reacts, Post, Reacts, User]$ (other users who react to the same post). *Relational variables* consist of a relational path and an attribute. For example, the relational variable $[User, Reacts, Post].Engagement$ corresponds to the overall engagements of the post that a user reacts to. The first item (i.e. $User$) in the relational path corresponds to the *perspective* of the relational variable. A terminal set, $P|_{i_k}$ is the terminal item on the relational path $P = [I_j, ..., I_k]$ consisting of instances of class $I_k \in \mathcal{E} \cup \mathcal{R}$.

A relational causal model $\mathcal{M} = \langle \mathcal{S}, \mathcal{D} \rangle$, is a collection of relational dependencies defined over schema $\mathcal{S}$. *Relational dependencies* consist of two relational variables, cause and effect. As an example, consider the following relational dependency $[Post, Reacts, User].Sentiment \rightarrow [Post].Engagement$ which states that the engagement of a post is affected by the actions of users who react on that post. In Figure 1, the arrows represents relational dependencies. Note that, all causal dependencies are defined with respect to a specific perspective.

### 3.5. Ground Graph and Abstract Ground Graph

A realization of a relational model $\mathcal{M}$ with a relational skeleton is referred to as the *ground graph* $GG_{\mathcal{M}}$. It is a directed graph consisting attributes of entities in the skeleton as nodes and relational dependencies among them as edges. A single relational model is actually a template for a set of possible ground graphs based on the given schema. A ground graph has the same semantic as a graphical model. Given a relational model $\mathcal{M}$ and a relational skeleton $s$, we can construct a ground graph $GG_{\mathcal{M}_s}$ by applying the relational dependencies as specified in the model to the specific instances of the relational skeleton. Figure 2b shows the ground graph for the relational model from

Figure 1. The relational dependencies present in the given RCM may temp one to conclude a conditional independence statement: $[User].Sentiment \perp\!\!\!\perp [Media].Preference|[Post].Engagement$. However, when the model is unrolled in a ground graph we see the corresponding statement is not true (i.e. $[Bob].Sentiment \not\perp\!\!\!\perp [M1].Preference|[P1].Engagement$) since there is an alternative path through $[Alice].Sentiment$ and $[P2].Engagement$ which is activated when conditioned on $[P1].Engagement$. This shows why generalization over all possible ground graphs is hard.

An *abstract ground graph* (AGG) is an abstract representation that solves the problem of generalization by capturing the consistent dependencies in all possible ground graphs and representing them as a directed graph. AGGs are defined for a specific perspective and *hop threshold*, $h$. Hop threshold refers to the maximum length of the relational paths allowed in a specific AGG. There are two types of nodes in AGG, relational variables and intersection variables. Intersection variables are constructed from pairs of relational variables with non-empty intersections (Maier et al., 2013b). For example, $[User, Reacts, Post]$ refers to the set of posts a user reacts to whereas $[User, Reacts, Post, Reacts, User, Reacts, Post]$ refers to the set of other posts reacted by other users who also reacted to the same post as the given user. These two sets of posts can overlap which is reflected by the corresponding intersection variable. Edges between a pair of nodes of AGG exist if the instantiations of those constituting relational variables contain a dependent pair in all ground graphs. We define $\overline{W}$ as the set of nodes augmented with their corresponding intersection nodes for the set of relational variables $\overline{W}$: $\overline{W} = W \cup \bigcup_{W \in W}\{W \cap W'|W \cap W'$ is an intersection node in $\text{AGG}_{\mathcal{M}_s}\}$. Figure 2c presents the AGG from the perspective of $User$ and with $h = 6$ corresponding to the model from Figure 1. The AGG shows that the sentiment of a user is no longer independent of media preference given just engagements of the corresponding posts the user reacts to. We also need to condition on the sentiment of other users who reacted to the same post.

### 3.6. Relational d-separation

Relational model describes a template for many possible instantiations of a relational schema. In order to reason about conditional independence facts entailed in all instances of a given relational template, Maier et al. (2013b) develop a relational counterpart for $d$-separation criteria. Two sets of relational variables $X$ and $Y$ from a given perspective are said to be $d$-separated by another set $Z$ if and only if the terminal sets of $X$ and $Y$ are $d$-separated by the terminal set of $Z$ from the given perspective in all possible ground graphs of the given model. Maier et al. (2013b) introduce AGG as a means to reason about relational $d$-separation queries from a given perspective. The soundness and completeness of relational $d$-separation for AGG relies on the following assumptions:

**A 1** *The relational model is acyclic.*

**A 2** *There are no unobserved confounders in the relational model.*

Here, soundness refers to the fact that any $d$-separation relationship found in AGG implies corresponding $d$-separation relationship in all ground graphs it represents whereas completeness claims that the $d$-separation facts that hold across all ground graphs are also entailed by $d$-separation on the AGG. The soundness of relational $d$-separation under AGG is already proved by Maier et al. (2013b). However, the conditions under which completeness holds have been an open question since Lee and Honavar (2015) show that the initial formulation of Maier et al. (2013b) is not complete. We resolve this question in Section 4.4 and show that AGG is also complete under certain realistic assumptions.

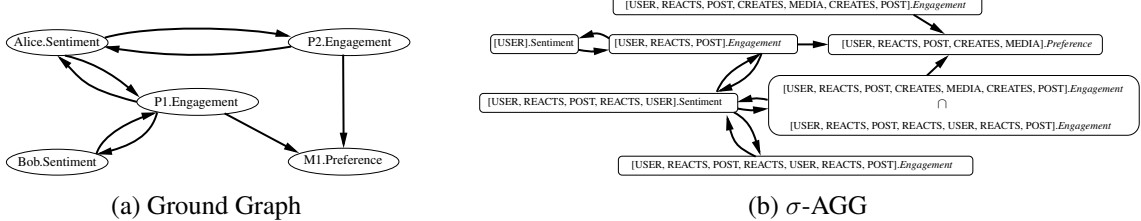

(a) Ground Graph    (b) $\sigma$-AGG

Figure 3: Ground graph and $\sigma$-AGG for the cyclic relational model shown in Figure 1.

## 4. Relational Systems With Cycles

In this section, we develop definitions for cyclic relational causal models (RCM) and an abstract representation that allows causal reasoning and discovery. We introduce a criterion for abstracting cyclic RCMs and provide proofs for its correctness.

### 4.1. Cyclic RCM

An RCM is cyclic when the set of relational dependencies form one or more arbitrary length cycles.

**Definition 4 (Cyclic RCM)** *A relational model $\mathcal{M} = (\mathcal{S}, \mathcal{D})$ is said to be cyclic if the set of relational dependencies $\mathcal{D}$ constructs one or more directed cycles of arbitrary length.*

Cycles in RCM represent equilibrium states among a set of nodes in the ground graphs. It implies that the ground graphs are no longer guaranteed to be DAGs. In order to facilitate this, we propose a revised definition of relational dependency provided by Maier et al. (2013b) by relaxing the restriction of having different attribute classes for cause and effect.

**Definition 5 (Relational Dependency)** *A relational dependency $[I_j, ..., I_k].X' \rightarrow [I_j].X$ is a directed probabilistic dependence from any attribute class $X'$ to $X$ through the relational path $[I_j, ..., I_k]$ such that $I_j, ..., I_k \in \mathcal{E} \cup \mathcal{R}$, $X, X' \in \mathcal{A}$. Note that it is possible to have $X = X'$.*

Figure 1 shows an example relational model with cyclic dependencies (i.e. dashed arrows). We see a pair of dependencies $[Post, Reacts, User].Sentiment \rightarrow [Post].Engagement$ and $[Post].Engagement \rightarrow [Post, Reacts, User].Sentiment$ which are inverse to each other and form a feedback loop. However, this mere feedback loop prohibits the use of AGG to answer relational causal query that asks whether a user's Sentiment about a post they reacted to is independent of the preference of media given the posts. Unfortunately, the work by Maier et al. (2013b) is not sufficient to reason about conditional independence relationships in the ground graphs of such relational models since it contain cycles. This motivates us to introduce a new criterion that enables the abstraction of relational queries with cyclic dependencies over all ground graphs.

### 4.2. Relational $\sigma$-separation

Conditional independence facts are only useful when they hold across all ground graphs that are consistent with the model. Maier et al. (2013b) show that relational $d$-separation is sufficient to achieve that for acyclic models. However, such abstraction is not possible for cyclic models since the correctness of $d$-separation is not guaranteed for cyclic graphical models (Spirtes, 1995; Neal,

2000). In this work, we propose the following definition of relational $\sigma$-separation specifically for cyclic relational models:

**Definition 6 (Relational $\sigma$-separation)** *Let $\boldsymbol{X}$, $\boldsymbol{Y}$, and $\boldsymbol{Z}$ be three distinct sets of relational variables with the same perspective $B \in \mathcal{E} \cup \mathcal{R}$ defined over relational schema $\mathcal{S}$. Then, for relational model structure $\mathcal{M}$, $\boldsymbol{X}$ and $\boldsymbol{Y}$ are $\sigma$-separated by $\boldsymbol{Z}$ if and only if, for all skeletons $s \in \sum_{\mathcal{S}}$, $\boldsymbol{X}|_b$ and $\boldsymbol{Y}|_b$ are $\sigma$-separated by $\boldsymbol{Z}|_b$ in ground graph $GG_{\mathcal{M}_s}$ for all instances $b \in s(B)$ where $s(B)$ refers to the instances of $B$ in skeleton $s$.*

The definition directly follows from the definition of relational $d$-separation. If there exists even one skeleton and faithful distribution represented by the relational model for which $\boldsymbol{X} \not\perp\!\!\!\perp \boldsymbol{Y}|\boldsymbol{Z}$, then $\boldsymbol{X}|_b$ and $\boldsymbol{Y}|_b$ are not $\sigma$-separated by $\boldsymbol{Z}|_b$ for $b \in s(B)$.

### 4.3. $\sigma$-Abstract Ground Graph

We refer to the lifted representation for cyclic RCMs as *$\sigma$-abstract ground graph* or *$\sigma$-AGG*. A $\sigma$-AGG is constructed using the same *extend* method used to construct AGG (Maier et al., 2013b).

**Definition 7 ($\sigma$-Abstract Ground Graph)** *An abstract ground graph $\sigma$-$AGG_{\mathcal{M}} = (V, E)$ for relational model structure $\mathcal{M} = (\mathcal{S}, \mathcal{D})$, perspective $B \in \mathcal{E} \cup \mathcal{R}$, and hop threshold $h \in \mathbb{N}^0$ is a directed graph that abstracts the dependencies $\mathcal{D}$ for all ground graphs $GG_{\mathcal{M}_s}$, where $s \in \sum_{\mathcal{S}}$. The $\sigma$-$AGG_{\mathcal{M}_s}$ is a directed cyclic graph with the following nodes and edges:*

1. *$V = RV \cup IV$, where*
    - *(a) $RV$ is the set of relational variables with a path of length at most $h + 1$.*
    - *(b) $IV$ are intersection variables between pairs of relational variables that could intersect*
2. *$E = RVE \cup IVE$, where*
    - *(a) $RVE \subset RV \times RV$ are the relational variable edges*
    - *(b) $IVE \subset (IV \times RV) \cup (RV \times IV)$ are the intersection variable edges. This is the set of edges that intersection variables "inherit" from the relational variables that they were created from*

Since the construction of an AGG and a $\sigma$-$AGG_{\mathcal{M}_s}$ is identical, they share mostly identical properties as defined by Maier et al. (2013b) for AGG. The main difference being the existence of cycles. Consequently, the goal of $\sigma$-$AGG_{\mathcal{M}_s}$ is to reason about relational $\sigma$-separation queries instead of relational $d$-separation. Figure 3b shows the $\sigma$-$AGG_{\mathcal{M}_s}$ corresponding to the cyclic RCM in Figure 1 with a pairwise feedback loop. It is similar to the AGG in Figure 2c but allows cycles without violating the conditional independence statements under $\sigma$-separation which are otherwise undefined with $d$-separation.

### 4.4. Soundness and Completeness of Relational $\sigma$-separation

In order to discuss soundness and completeness of relational $\sigma$-separation we first address the open problem of necessary conditions for the completeness of relational $d$-separation. Previous work has shown that the original claim of completeness of relational $d$-separation by Maier et al. (2013a)

cannot be guaranteed for any relational model (Lee and Honavar, 2015). A counterexample has been developed as well. In this work, we show that relational $d$-separation is complete under the following assumption:

**A 3** *The degree of any entity in the relational skeleton is greater than 1.*

Note that, this assumption is about the topology of the ground graphs, i.e., the network which defines how entities are connected to each other. It only allows entities that are connected to at least two other entities. For example, if the entities are users in a social network, the framework would only consider users who have degree at least two, i.e., are connected to at least two other users. While this restricts the space of graph topologies allowable under the results in this work, many networks observed in real world domains, such as social networks, have minimum degree greater than one. We introduce the following lemma that establishes sufficient conditions for AGGs to be realizable in ground graphs. This result may be of independent interest, since it provides sufficient conditions for soundness in the original presentation of relational $d$-separation under additional assumptions (Maier et al., 2013b; Lee and Honavar, 2015).

**Lemma 1** *Under assumption 3, every abstract ground graph can be realized as a ground graph. That is, for every acyclic relational model $\mathcal{M}$ and skeleton $s \in \sum_{\mathcal{S}}$ any relational variable in $AGG_{\mathcal{M}_s}$ has non-empty terminal sets in some ground graph $GG_{\mathcal{M}_s}$.*

**Proof**

We first consider the conditions under which empty terminal sets can occur, resulting in an AGG that is unrealizable in the ground graphs. There are two necessary and sufficient conditions for empty terminal sets to appear in all ground graphs corresponding to an AGG. First, there must be at least one intersection variable present in the AGG. If no intersection variable exist in the AGG, then the completeness proof of relational $d$-separation by Maier et al. (2013b) holds. The second condition is that the intersection must be on a path consisting of only one-to-one relationships. In order to understand this condition, lets look at an example with the following relational paths from a hypothetical relational model which is a generalization of the counterexample given by Lee and Honavar (2015) [2]:

- $P = [E_i, \ldots, R_j, \ldots, E_k]$
- $S = [E_i, \ldots, R_j, \ldots, R_m, \ldots, E_s]$
- $Q = [E_i, \ldots, R_j, \ldots, R_m, E_k, \ldots, E_q]$
- $S' = P + [R_m, \ldots, E_s]$

where $E_i, E_k, E_q, E_s$ are some entity classes, $R_j, R_m$ are relationship classes, "..." are arbitrary valid sequences of entities and relationships, and $+$ represents the concatenation of relational paths. Let's assume two relational dependencies exist in the given model, $P.X \rightarrow Q.Y$ and $S.Z \rightarrow Q.Y$ where $X, Y, Z$ are attributes of corresponding entity classes. By definition, the corresponding edges $P.X \rightarrow Q.Y, S.Z \rightarrow Q.Y$ appear in the AGG. Since $S$ and $S'$ are intersectable an additional edge $S.Z \cap S'.Z \rightarrow Q.Y$ also appears in the AGG. Such a model can be realized in many possible ground graphs. However, if we restrict the relationships to be strictly one-to-one, then there is only one skeleton structure possible to satisfy the relational dependencies at the cost of $S'.Z$ having empty

---

2. The complete counterexample and figure explaining it is given in the Appendix

terminal sets since an intance of $R_m$ can connect to only one instance of $E_k$. If we allow many-to-many relationships then we can always construct a skeleton where an instance of $R_m$ connects to two instances of $E_k$ to produce non-empty terminal sets for both $Q$ and $S'$.

Since assumption 3 prohibits the second condition, it essentially implies that any relational variable in $\text{AGG}_{\mathcal{M}_s}$ results into non-empty terminal sets in corresponding ground graphs for every acyclic relational model $\mathcal{M}$ and skeleton $s \in \sum_{\mathcal{S}}$ which completes the proof for Lemma 1. ∎

The following proposition establishes the completeness of AGG for relational $d$-separation under the assumption of a minimum degree greater than 1.

**Proposition 1** *AGG is sound and complete for relational $d$-separation under assumption 3.*

**Proof** Following lemma 1, the original proof of soundness and completeness of relational $d$-separation by Maier et al. (2013b) directly applies which proves proposition 1. ∎

The correctness of our approach to relational $\sigma$-separation relies on several facts which are similar to the case for AGG: (1) $\sigma$-separation is valid for directed cyclic graphs; (2) ground graphs are directed cyclic graphs; and (3) $\sigma$-AGGs are directed cyclic graphs that represent exactly the edges that could appear in all possible ground graphs. Note that we no longer need assumption 1, but assumptions 2 and 3 are adopted from relational $d$-separation. Using the previous definitions and lemmas, the following additional assumptions and sequence of results proves the correctness of our approach to identifying independence in cyclic relational models.

**A 4** *The given cyclic relational model structure is $\sigma$-faithful.*

**Theorem 1** *The rules of $\sigma$-separation are sound and complete for cyclic directed graphs.*

**Proof** Forré and Mooij (2017) show that for quite general structural equation models HEDGes[3] always follow a directed global Markov property based on $\sigma$-separation which completes the proof for soundness since directed cyclic graphs are subsets of HEDGes. The completeness claim is already covered by Assumption 4. ∎

**Theorem 2** *For every cyclic RCM $\mathcal{M} = (\mathcal{S}, \mathcal{D})$ and skeleton $s \in \sum_{\mathcal{S}}$ such that relational variables involved in $\mathcal{D}$ are non-empty, the ground graph $GG_{\mathcal{M}_s}$ is a cyclic directed graph.*

**Proof**

Let's assume for contradiction that there exists an acyclic ground graph $g$ which is a realization of a given cyclic RCM $\mathcal{M} = (\mathcal{S}, \mathcal{D})$ and skeleton $s \in \sum_{\mathcal{S}}$. According to the definition of ground graphs, the edges of ground graphs are directly constructed based on the relational dependencies of the model. Definition 4 states that a cyclic RCM consists of one or more cycles formed by the relational dependencies. Assume a cycle in cyclic RCM is formed by the pair of relational dependencies as follows: a) $P.j \rightarrow Q.k$, and b) $Q.k \rightarrow P.j$ where $P$ and $Q$ are relational paths from some perspective $b$ and $i, j$ refers to two attribute classes. By construction of $g$ there must be two nodes $a, b$ in $g$ corresponding to $P.j$ and $Q.k$ respectively. Moreover, the definition of $g$ requires two edges $a \rightarrow b$ and $b \rightarrow a$ to be present in the ground graph. But such edges constructs a cycle which is contradictory to the initial claim. Thus, the ground graph $g$ must be cyclic. ∎

---

3. The definition of HEDG is provided in the Appendix

**Theorem 3** *For every cyclic relational model structure $\mathcal{M}$ and perspective $B \in \mathcal{E} \cup \mathcal{R}$, the $\sigma$-AGG$_{\mathcal{M}_s}$ is sound and complete for all ground graphs $GG_{\mathcal{M}_s}$ with skeleton $s \in \sum_{\mathcal{S}}$.*

The proof follows from the proof of soundness and completeness of AGG (Maier et al., 2013b). The complete proof is provided in the Appendix.

**Theorem 4** *The abstract ground graph $\sigma$-AGG$_{\mathcal{M}_s}$ is a cyclic directed graph if and only if the underlying relational model structure is cyclic.*

**Proof**

Let $\mathcal{M}$ be an arbitrary (possibly) cyclic relational model structure, and let $B \in \mathcal{E} \cup \mathcal{R}$ be an arbitrary perspective. It is clear by Definition 7 that every edge in the abstract ground graph $\sigma$-AGG$_{\mathcal{M}_s}$ is directed by construction. Assume for contradiction that no cycles exists in $\sigma$-AGG$_{\mathcal{M}_s}$ even if the relational dependencies form one or more cycles. Now assume the following two dependencies are part of the given relational model $\mathcal{M}$: 1. $[I_j, ..., I_k].X \rightarrow [I_j].Y \in D$, 2. $[I_j].Y \rightarrow [I_j, ..., I_k].X \in D$ where $I_j, ..., I_k \in \mathcal{E} \cup \mathcal{R}$. By Definition 7, all edges inserted in $\sigma$-AGG$_{\mathcal{M}_s}$ are drawn from some dependency in $\mathcal{M}$ and edges in $\sigma$-AGG$_{\mathcal{M}_s}$ are constructed for all the dependencies in $D$. As a result there must be corresponding edges in the $\sigma$-AGG$_{\mathcal{M}_s}$ for both dependencies that form a cycle, which contradicts the assumption.

Now, assume that a $\sigma$-AGG$_{\mathcal{M}_s}$ is acyclic even if the underlying RCM is cyclic. Using the same argument as above we can say that the edges in the $\sigma$-AGG$_{\mathcal{M}_s}$ constructed based on the dependencies in $\mathcal{D}$. If a cycle exists in the $\sigma$-AGG$_{\mathcal{M}_s}$ it directly implies the existence of a cycle in the RCM which leads to a contradiction. Thus the proof completes from both directions. ∎

**Theorem 5** *Relational $\sigma$-separation is sound and complete for $\sigma$-AGG. Let $\mathcal{M}$ be a (possibly) cyclic relational model structure, and let $\mathbf{X}$, $\mathbf{Y}$, and $\mathbf{Z}$ be three distinct sets of relational variables defined over relational schema $\mathcal{S}$. Then, $\overline{\mathbf{X}}$ and $\overline{\mathbf{Y}}$ are $\sigma$-separated by $\overline{\mathbf{Z}}$ on the abstract ground graph $\sigma$-AGG$_{\mathcal{M}_s}$ if and only if for all skeletons $s \in \sum_{\mathcal{S}}$ and for all perspectives $b \in s(B)$, $\mathbf{X}|_b$ and $\mathbf{Y}|_b$ are $\sigma$-separated by $\mathbf{Z}|_b$ in ground graph $GG_{\mathcal{M}_s}$.*

**Proof**

We must show that $\sigma$-separation on an abstract ground graph implies $\sigma$-separation on all ground graphs it represents (soundness) and that $\sigma$-separation facts that hold across all ground graphs are also entailed by $\sigma$-separation on the abstract ground graph (completeness). The proof follows from the proof of soundness and completeness of AGG (Maier et al., 2013b).

**Soundness:**

Assume that $\overline{\mathbf{X}}$ and $\overline{\mathbf{Y}}$ are $\sigma$-separated by $\overline{\mathbf{Z}}$ on $\sigma$-AGG$_{\mathcal{M}_s}$. Assume for contradiction that there exists an item instance $b$ such that $\mathbf{X}|_b$ and $\mathbf{Y}|_b$ are not $\sigma$-separated by $\mathbf{Z}|_b$ in the ground graph $GG_{\mathcal{M}_s}$ for some arbitrary skeleton $s$. Then, there must exist a $\sigma$-connecting path $p$ from some $x \in \overline{\mathbf{X}}|_b$ to some $y \in \overline{\mathbf{Y}}|_b$ given all $z \in \overline{\mathbf{Z}}|_b$. By Theorem 3, $\sigma$-AGG$_{\mathcal{M}_s}$ is complete, so all edges in $GG_{\mathcal{M}_s}$ are captured by edges in $\sigma$-AGG$_{\mathcal{M}_s}$. So, path $p$ must be represented from some node in $\{N_x | x \in N_x|_b\}$ to some node in $\{N_y | y \in N_y|_b\}$, where $N_x$, $N_y$ are nodes in $\sigma$-AGG$_{\mathcal{M}_s}$. If $p$ is $\sigma$-connecting in $GG_{\mathcal{M}_s}$, then it is $\sigma$-connecting in $\sigma$-AGG$_{\mathcal{M}_s}$, implying that $\overline{\mathbf{X}}$ and $\overline{\mathbf{Y}}$ are not $\sigma$-separated by $\overline{\mathbf{Z}}$. So, $\mathbf{X}|_b$ and $\mathbf{Y}|_b$ must be $\sigma$-separated by $\mathbf{Z}|_b$.

**Completeness:**

Assume that $\boldsymbol{X}|_b$ and $\boldsymbol{Y}|_b$ are $\sigma$-separated by $\boldsymbol{Z}|_b$ in the ground graph $GG_{\mathcal{M}_s}$ for all skeletons $s$ for all $b \in s(B)$. Assume for contradiction that $\bar{\boldsymbol{X}}$ and $\bar{\boldsymbol{Y}}$ are not $\sigma$-separated by $\bar{\boldsymbol{Z}}$ on $\sigma$-AGG$_{\mathcal{M}_s}$. Then, there must exist a $\sigma$-connecting path $p$ for some relational variable $X \in \bar{\boldsymbol{X}}$ to some $Y \in \bar{\boldsymbol{Y}}$ given all $Z \in \bar{\boldsymbol{Z}}$. By Theorem 3, $\sigma$-AGG$_{\mathcal{M}_s}$ is sound, so every edge in $\sigma$-AGG$_{\mathcal{M}_s}$ must correspond to some pair of variables in some ground graph. So, if $p$ is $\sigma$-connecting in $\sigma$-AGG$_{\mathcal{M}_s}$, then there must exist some skeleton $s$ such that $p$ is $\sigma$-connecting in $GG_{\mathcal{M}_s}$ for some $b \in s(B)$, implying that $\sigma$-separation does not hold for that ground graph. So, $\bar{\boldsymbol{X}}$ and $\bar{\boldsymbol{Y}}$ must be $\sigma$-separated by $\bar{\boldsymbol{Z}}$ on $\sigma$-AGG$_{\mathcal{M}_s}$. ∎

Maier et al. (2013b) show that relational $d$-separation is equivalent to the Markov condition on acyclic relational models. However, it doesn't hold for cyclic relational model. Here, we show how relational $\sigma$-separation is equivalent to the Markov condition on cyclic relational models.

**Definition 8 (Relational $\sigma$-separation Markov Condition)** *Let $X, Y, Z$ be relational variables for perspective $B \in \mathcal{E} \cup \mathcal{R}$ defined over relational schema $\mathcal{S}$. For any solution $(\boldsymbol{\mathcal{X}}, \boldsymbol{\epsilon})$ of a relational model $\mathcal{M}$ which follows a simple SCM,*

$$X \underset{\mathcal{M}}{\overset{\sigma}{\perp\!\!\!\perp}} Y | Z \implies \boldsymbol{\mathcal{X}}_X \underset{\mathbb{P}_{\mathcal{M}}(\boldsymbol{\mathcal{X}})}{\perp\!\!\!\perp} \boldsymbol{\mathcal{X}}_Y | \boldsymbol{\mathcal{X}}_Z, \;\; \text{if and only if}$$

$$x \underset{GG_{\mathcal{M}}}{\overset{\sigma}{\perp\!\!\!\perp}} y | z \implies \boldsymbol{\mathcal{X}}'_x \underset{\mathbb{P}_{GG_{\mathcal{M}}}(\boldsymbol{\mathcal{X}}')}{\perp\!\!\!\perp} \boldsymbol{\mathcal{X}}'_y | \boldsymbol{\mathcal{X}}'_z, \;\; \text{for } \forall x \in X|_b, \; \forall y \in Y|_b, \; \forall z \in Z|_b$$

*in ground graph $GG_{\mathcal{M}_s}$ for all skeletons $s \in \sum_{\mathcal{S}}$ and for all $b \in s(B)$ where $(\boldsymbol{\mathcal{X}}', \boldsymbol{\epsilon}')$ refers to the solution of the SCM corresponding to the ground graphs.*

In other words, $\sigma$-separation of two relational variables $\boldsymbol{X}$ and $\boldsymbol{Y}$ given a third relational variable $\boldsymbol{Z}$ would imply $\boldsymbol{X}$ and $\boldsymbol{Y}$ are conditionally independent given $\boldsymbol{Z}$ if and only if, for all instances of $\boldsymbol{X}, \boldsymbol{Y}, \boldsymbol{Z}$ in all possible ground graphs, the same condition holds. Since ground graphs of cyclic RCM are directed cyclic graphs and $\sigma$-separation on $\sigma$-AGG$_{\mathcal{M}_s}$ is sound and complete (by Theorem 5), we can conclude that relational $\sigma$-separation is equivalent to the relational Markov property.

## 5. Conclusion

Cycles or feedback loops are common elements of many real-world system. Unfortunately, it is hardly studied in the field of causal inference primarily because it breaks the nice properties of directed acyclic graphs. As a result, cycles and feedback loops are mostly avoided in the domain of relational causal model. In this study, we take a step forward to bridge this gap by developing an abstract representation and a criterion to reason about statistical relationships in relational models with or without cycles under a general framework. We show that the new criterion called $\sigma$-separation can consistently capture the statistical independence relationships of all possible instantiations of a relational causal model. We believe that this work will open the door for further development including but not limited to causal structure learning of relational models with cycles.

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

## Appendix A. Preliminaries

### A.1. Directed Graphs with Hyperedges (HEDGes)

A directed graph with hyperedges or hyperedged directed graph (HEDG) is a tuple $G = (\mathcal{V}, \mathcal{E}, \mathcal{H})$, where $(\mathcal{V}, \mathcal{E})$ is a directed graph (with or without cycles) and $\mathcal{H}$ a simplicial complex over the set of vertices $\mathcal{V}$ of $\mathcal{G}$. A simplicial complex $\mathcal{H}$ over $\mathcal{V}$ is a set of subsets of $\mathcal{V}$ such that: 1) all single element sets $\{v\}$ are in $\mathcal{H}$ for $v \in \mathcal{V}$, and 2) if $F \in \mathcal{H}$ then also all subsets $F' \subseteq F$ are elements of $\mathcal{V}$.

The general directed global Markov property (gdGMP) for the HEDGes is stated as follows:

**Definition 9 (gdGMP)** *For all subsets $X, Y, Z \subseteq V$ we have the implication:*

$$X \overset{\sigma}{\underset{G}{\perp\!\!\!\perp}} Y | Z \implies X \underset{P_v}{\perp\!\!\!\perp} Y | Z$$

### A.2. Counterexample by Lee and Honavar (2015)

The following counterexample shows that AGG is not complete for relational $d$-separation.

**Example.** Let $\mathcal{S} = \langle \mathcal{E}, \mathcal{R}, \mathcal{A}, card \rangle$ be a relational schema such that: $\mathcal{E} = \{E_i\}_{i=1}^5$; $\mathcal{R} = \{R_j\}_{j=1}^3$ with $R_1 = \langle E_1, E_2, E_4 \rangle$, $R_2 = \langle E_2, E_3 \rangle$, $R_3 = \langle E_3, E_4, E_5 \rangle$; $\mathcal{A} = \{E_2 : \{Y\}, E_3 : \{X\}, E_5 : \{Z\}\}$; and $\forall_{R \in \mathcal{R}} \forall_{E \in \mathcal{E}} \, card(R, E) = one$. Let $\mathcal{M} = \langle \mathcal{S}, \mathcal{D} \rangle$ be a relational model with

$$\mathcal{D} = \{D_1.X \to [I_Y].Y, D_2.Z \to [I_Y].Y\}$$

such that $D_1 = [E_2, R_2, E_3, R_3, E_4, R_1, E_2, R_2, E_3]$ and $D_2 = [E_2, R_2, E_3, R_3, E_5]$. Let $P.X, Q.Y, S.Z, S'.Z$ be four relational variables of the same perspective $B = E_1$ where their relational paths are distinct where

- $P = [E_1, R_1, E_2, R_2, E_3]$  • $Q = [E_1, R_1, E_4, R_3, E_3, R_2, E_2]$
- $S = [E_1, R_1, E_4, R_3, E_5]$  • $S' = [E_1, R_1, E_2, R_2, E_3, R_3, E_5]$

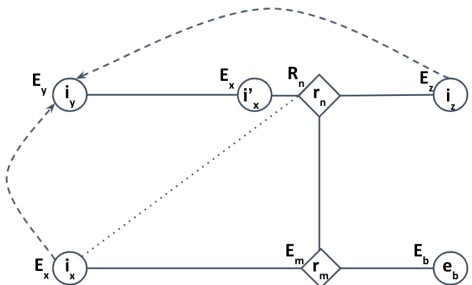

Figure 4: Construction of the counterexample by Lee and Honavar (2015). The notations inside the circles/rhombus refer to instances of the corresponding entity/relationship classes which are mentioned outside the shapes. The dashed lines represent relational dependencies. The dotted line represents a hypothetical connection that can nullify the counterexample under assumption 3.

Given the above example, Lee and Honavar (2015) make two claims:

*Claim 1.* $(\overline{P.X} \not\perp \overline{S'.Z} | \overline{Q.Y})_{AGG_{\mathcal{M}}}$

*Claim 2.* There is no $s \in \sum_{\mathcal{S}}$ and $b \in s(B)$ such that $(P.X|_b \not\perp S'.Z|_b | Q.Y|_b)_{GG_{\mathcal{M}_s}}$

Figure 4 shows the general pattern discussed in Section 4.4 regarding the construction of the counterexample by Lee and Honavar (2015).

## Appendix B. Proofs

The proof for Theorem 3 is given as follows:

**Proof**

Let $\mathcal{M} = (\mathcal{S}, \mathcal{D})$ be an arbitrary cyclic relational model structure and $B \in \mathcal{E} \cup \mathcal{R}$ an arbitrary perspective.

**Soundness:** To prove that $\sigma\text{-}AGG_{\mathcal{M}_s}$ is sound, we must show that for every edge $P_k.X \to P_j.Y$ in $\sigma\text{-}AGG_{\mathcal{M}_s}$, there exists a corresponding edge $i_k.X \to i_j.Y$ in the ground graph $GG_{\mathcal{M}_s}$ for some skeleton $s \in \sum_{\mathcal{S}}$, where $i_k \in P_k|_b$ and $i_j \in P_j|_b$ for some $b \in s(B)$. There are three subcases, one for each type of edge in an abstract ground graph:

(a) Let $[B, ..., I_k].X \to [B, ..., I_j].Y \in RVE$ be an arbitrary edge in $\sigma\text{-}AGG_{\mathcal{M}_s}$ between a pair of relational variables. Assume for contradiction that there exists no edge $i_k.X \to i_j.Y$ in any ground graph:

$$\forall s \in \Sigma_{\mathcal{S}}, \forall i_k \in [B, ..., I_k]|_b, \forall i_j \in [B, ..., I_j]|_b$$
$$(i_k.X \to i_j.Y \notin GG_{\mathcal{M_S}})$$

By Definition 7 for $\sigma\text{-}AGG_{\mathcal{M}_s}$, if $[B, ..., I_k].X \to [B, ..., I_j].Y \in RVE$, then the model must have dependency $[I_j, ..., I_k].X \to [I_j].Y \in \mathcal{D}$ such that $[B, ..., I_k] \in extend([B, ..., I_j], [I_j, ..., I_k])$. So, by the definition of ground graphs, there is an edge from every $i_k.X$ to every $i_j.Y$, where $i_k$ is in the terminal set for $i_j$ along $[I_j, ..., I_k]$. Therefore, there exists a ground graph $GG_{\mathcal{M}_s}$ such that $i_k.X \to i_j.Y \in GG_{\mathcal{M}_s}$, which contradicts the assumption.

(b) Let $P_1.X \cap P_2.X \to [B, ..., I_j].Y \in IVE$ be an arbitrary edge in $\sigma\text{-}AGG_{\mathcal{M}_s}$ between an intersection variable and a relational variable, where $P_1 = [B, ..., I_m, ..., I_k]$ and $P_2 = [B, ..., I_n, ..., I_k]$ with $I_m \neq I_n$. By Definition 7, if the $\sigma$-abstract ground graph has edge $P_1.X \cap P_2.X \to [B, ..., I_j].Y \in IVE$, then either $P_1.X \to [B, ..., I_j].Y \in RVE$ or $P_2.X \to [B, ..., I_j].Y \in RVE$. Then, as shown in case (a), there exists an $i_j \in [B, ..., I_j]|_b$ such that $i_k.X \to i_j.Y \in GG_{\mathcal{M}_s}$, which contradicts the assumption.

(c) Let $[B, ..., I_k].Y \to P_1.X \cap P_2.X \in IVE$ be an arbitrary edge in $\sigma\text{-}AGG_{\mathcal{M}_s}$ between an intersection variable and a relational variable, where $P_1 = [B, ..., I_m, ..., I_j]$ and $P_2 = [B, ..., I_n, ..., I_j]$ with $I_m \neq I_n$. The proof follows case (b) to show that there exists a skeleton $s \in \sum_{\mathcal{S}}$ and $b \in s(B)$ such that for all $i_k \in [B, ..., I_k]|_b$ there exists an $i_j \in P_1 \cap P_2|_b$ such that $i_k.X \to i_j.Y \in GG_{\mathcal{M}_s}$.

**Completeness:** To prove that the $\sigma$-abstract ground graph $\sigma\text{-}AGG_{\mathcal{M}_s}$ is complete, we show that for every edge $i_k.X \to i_j.Y$ in every ground graph $GG_{\mathcal{M}_s}$ where $s \in \sum_{\mathcal{S}}$, there is a set of corresponding edges in $\sigma\text{-}AGG_{\mathcal{M}_s}$. Specifically, the edge $i_k.X \to i_j.Y$ yields two sets of relational variables for some $b \in s(B)$, namely $\boldsymbol{P_k.X} = \{P_k.X | i_k \in P_k|_b\}$ and $\boldsymbol{P_j.Y} = \{P_j.Y | i_j \in P_j|_b\}$. Note that all relational variables in both $\boldsymbol{P_k.X}$ and $\boldsymbol{P_j.Y}$ are nodes in $\sigma\text{-}AGG_{\mathcal{M}_s}$, as are all pairwise intersection variables. We show that for all $P_k.X \in \boldsymbol{P_k.X}$ and for all $P_j.Y \in \boldsymbol{P_j.Y}$ either (a)

$P_k.X \rightarrow P_j.Y \in \sigma\text{-AGG}_{\mathcal{M}_s}$ (b) $P_k.X \cap P'_k.X \rightarrow P_j.Y \in \sigma\text{-AGG}_{\mathcal{M}_s}$ where $P'_k.X \in \boldsymbol{P_k.X}$, or (c) $P_k.X \rightarrow P_j.Y \cap P'_j.Y \in \sigma\text{-AGG}_{\mathcal{M}_s}$ where $P'_j.Y \in \boldsymbol{P_j.Y}$.

Let $s \in \sum_{\mathcal{S}}$ be an arbitrary skeleton, let $i_k.X \rightarrow i_j.Y \in GG_{\mathcal{M}_s}$ be an arbitrary edge drawn from $[I_j, ..., I_k].X \rightarrow [I_j].Y \in \mathcal{D}$, and let $P_k.X \in P_k.X, P_j.Y \in P_j.Y$ be an arbitrary pair of relational variables.

(a) If $P_k \in extend(P_j, [I_j, ..., I_k])$, then $P_k.X \rightarrow P_j.Y \in \sigma\text{-AGG}_{\mathcal{M}_s}$ by Definition 7.

(b) If $P_k \notin extend(P_j, [I_j, ..., I_k])$, but $\exists P'_k \in extend(P_j, [I_j, ..., I_k])$ such that $P'_k.X \in P_k.X$, then $P'_k.X \rightarrow P_j.Y \in \sigma\text{-AGG}_{\mathcal{M}_s}$, and $P_k.X \cap P'_k.X \rightarrow P_j.Y \in \sigma\text{-AGG}_{\mathcal{M}_s}$ by Definition 7.

(c) If $\forall P \in extend(P_j, [I_j, ..., I_k])(P.X \notin Pk.X)$, then $\exists P'_j$ such that $i_j \in P'_j|_b$ and $P_k \in extend(P'_j, [I_j, ..., I_k])$. Therefore, $P'_j.Y \in P_j.Y, P_k.X \rightarrow P'_j.Y \in \sigma\text{-AGG}_{\mathcal{M}_s}$, and $P_k.X \rightarrow P'_j.Y \cap P_j.Y \in \sigma\text{-AGG}_{\mathcal{M}_s}$ by Definition 7.

■

