# OpenReview forum: "Relational Causal Models with Cycles: Representation and Reasoning"
_cclear.cc/CLeaR/2022/Conference — CLeaR 2022 Poster_

### Official Review · Reviewer_Wvcq · 2021-11-22

**Confidence:** 2
**Overall Score:** 4

**Main Review:**

The most important problem with this manuscript is its lack of clarity. The concepts on which it builds, i.e. relational causal models and cyclic causal models, are non-standard and thus should be defined clearly to help the non-expert reader (like myself) undertand the contributions. I often had to refer to either Forré and Mooij (2017) and Maier et al. (2013b) to understand some notions specifically related to cyclic causal models and relational causal models.. This is not necessarily a bad thing given some notions are fairly complex and cannot be fully explained in the background section of a conference paper, but, once I understood the notion, it became clear that it could have been explained more clearly in the manuscript without eating up too much space. This lack of clarity prevents me from recommending acceptance at the moment.

### Originality:
To the best of my knowledge, cyclic relational causal models are novel and were not investigated before. That being said, most theoretical results in this paper seem to be rather direct applications of the results of Forré and Mooij (2017) and Maier et al. (2013b), which leads me to think this work has medium originality.

### Significance:
The motivation for relational causal models was almost completely missing from the manuscript. In addition, the cycle between "employee.performance" and "product.quality" in the motivating example is never explained. As a result, the overall contribution of this work lacked motivation which led me to believe its significance is somewhere between low and medium.

### Clarity:
- Related work is hard to follow for someone unfamiliar with relational causal models and cyclic models. Consider integrating it with the Preliminaries section.
- Definition 1: should be d-separation instead of sigma-separation.
- Section 3.4: This section was very hard to understand. Why are they more than one relational skeleton? The notion of "relational path" and "perspective" feels largely unmotivated. What is the difference between [Alice].salary and [Laptop, develops, Alice].salary?
- In Figure 2, it would be useful to have a relational skeleton represented between the relational causal model and the ground graph.
- The explanation of an "abstract ground graph" (AGG) in Section 3.5 was very fast given its central importance. After reading about it in Maier et al (2013b), I noticed that the definition in the present manuscript does not mention that the AGG is actually dependent on the perspective and the "hop threshold" h. This seems like an important point completely missed. This omission makes the relational counterpart for graphical d-separation in section 3.6 very confusing. Also, the notion of "intersection variables", which is important for later results, is very poorly explained. What does the reader should understand from "Intersection variables are constructed from pairs of relational variables with non-empty intersections."? The intersection of relational variables was never defined.
- Definition 5 what is s(B)? Couldn't find the definition.
- Definition 6: the hop threshold "h" appears for the first time next to a AGG, without any prior explanation. Also, points 1.b and 2.b are not clearly explained. The former refers "intersection variables" which were ambiguously defined and the latter uses "inherit" in a very vague sense.
- Shouldn't Definition 7 be a theorem or a proposition?


### Technical quality:
- I am not familiar with cyclic causal models, but the paragraph starting with "Spirtes (1995); Neal (2000)" just after definition 1 was confusing. It says that d-separation are not sufficient to entail all the corresponding conditional independence relations in a direct cyclic graph. However, this statement is meaningless without formalizing a connection between the causal graph and the distribution via a Markov property or a structural causal model. For instance, in an acyclic causal model, we can assume that the distribution is entailed by an SCM with some graph G. Then, it is meaningful to talk about the relation between the graph G and the conditional independences in the distribution.
- On a related note, it is mentioned nowhere how one should translate the cyclic graph into conditional independence statements. I suspect this is what Definition 2 is for: \sigma-separation implies corresponding conditional independence? It should be explicitly stated.
- The lack of clarity made reading the main theoretical contributions of this work very difficult. As a consequence, I could not assess their quality.

# Response to rebuttal
I thank the authors for providing clarifications. I maintain my initial assessment: the lack clarity and motivation are the two main problems with this manuscript that prevent me from recommending acceptance at the moment. I encourage the authors to improve on these aspects in a future iteration of their work.

**Summary:**

This work generalizes the relational causal models to account for cycles by introducing "relational sigma-separation" based on the work of Forré and Mooij (2017).

---

> ### Author Response · Authors · 2021-12-04
> **Response to reviewer Wvcq (Part 2/2)**
>
> > In Figure 2, it would be useful to have a relational skeleton represented between the relational causal model and the ground graph.
>
> Thank you for this suggestion, it makes sense and we can include it.
>
> > Present manuscript does not mention that the AGG is actually dependent on the perspective and the "hop threshold" h. This seems like an important point completely missed.
>
> We thank the reviewer for pointing this out. We mention perspective at the end of section 3.4 and hop threshold at point 1a of definition 6 but if the AGG’s dependence on them was not clear, then we will definitely make this more explicit.
>
> > What should the reader understand from "Intersection variables are constructed from pairs of relational variables with non-empty intersections."?
>
> Intersection variables refer to the result of set intersection on multiple relational variables. For example, [Laptop, Develop, Employee] refers to the set of employees who develop the product laptop. On the other hand, [Keyboard, Develop, Employee] refers to the set of employees who develop the product Keyboard. If we consider the intersection of these two sets we get the employees who develop both laptop and keyboard. Such intersection variables are a key part of AGG. Since we directly use the notion from Maier et. al. we didn’t include a lot of detail on them in the paper due to space constraints. We plan to include a section in the appendix with a detailed description of intersection variables.
>
> > Definition 5 what is s(B)? Couldn't find the definition.
>
> s(B) refers to the entity instances of entity type B in the given skeleton s. We will clarify this notation.
>
> > Shouldn't Definition 7 be a theorem or a proposition?
>
> Yes, it should be a theorem. That was an unintentional mistake that we will correct.
>
> > The paragraph starting with "Spirtes (1995); Neal (2000)" just after definition 1 was confusing. It says that d-separation are not sufficient to entail all the corresponding conditional independence relations in a direct cyclic graph. However, this statement is meaningless without formalizing a connection between the causal graph and the distribution via a Markov property or a structural causal model.
>
> We will add a sentence in the beginning of that paragraph to formalize that connection. The statement itself is an established result from the works of Spirtes (1995); Neal (2000).
>
> > It is mentioned nowhere how one should translate the cyclic graph into conditional independence statements. I suspect this is what Definition 2 is for: $\sigma$-separation implies corresponding conditional independence? It should be explicitly stated.
>
> The conditional independencies in a cyclic graph are similar to the acyclic case except that one has to account for the fact that a node can serve both as a collider and non-collider between two other nodes in the presence of cycles. For example, not observing B in the graph B->A->B<-C makes A and C dependent because of the chain C->B->A despite B being a collider in A->B<-C. \$sigma$-separation formalizes how these independences can be found. We will clarify this further.

---

> ### Author Response · Authors · 2021-12-04
> **Response to reviewer Wvcq (Part 1/2)**
>
> We thank the reviewer for the feedback and provide responses to each point.
>
> > Most theoretical results in this paper seem to be rather direct applications of the results of Forré and Mooij (2017) and Maier et al. (2013b), which leads me to think this work has medium originality.
>
> While we agree that our work builds upon the works of Forré and Mooij (2017) and Maier et al. (2013b), neither one considers the implications of introducing cycles in relational models. The contributions of our work include the following:
> 1) To the best of our knowledge, this is the only work that aims to reason about cyclic relational causal models. While much progress has been made on representation and learning of RCMs, none of them address feedback loops and do not allow for reasoning about identification in the presence of feedback loops. We study cyclic RCMs since they offer the necessary representation to reason about a lot of real-world causal problems where SUTVA does not hold.
> 2) We introduce a new abstract representation ($\sigma$-AGG) for cyclic RCMs and a means (relational $\sigma$-separation) to  reason about relational queries on such models. We provide theoretical justification for the proposed representation.
> 3) We provide sufficient conditions for the completeness of AGG which resolves the counterexample presented in (Lee et. al. UAI 2015). This non-trivial contribution greatly improves the applicability of AGG and has implications beyond the scope of cyclic relational models.
>
> > The motivation for relational causal models was almost completely missing from the manuscript. In addition, the cycle between "employee.performance" and "product.quality" in the motivating example is never explained.
>
> The main motivation for our work is to enable the study of social phenomena through rigorous reasoning and identification of effects of interest (e.g., contagion). Some of the literature on interference and SCM considers only templates with pairwise relationships (E.g., Ogburn & VanderWeele 2014, Shalizi & Thomas 2011). This is unrealistic because social networks are not composed of pairs of nodes that are independent of other pairs of nodes. To be realistic, these templates would need to have dependencies with themselves. RCMs allow us to represent such more complex dependencies. What RCMs offer in this context is two-fold: 1) they offer a general expressive language to reason about causal dependencies in relational data where a single template can represent infinite instances. 2) They allow a rich set of tools (i.e. relational d-separation, AGG) which enable representation and learning of relational causal models from observational samples.
>
> In the given hypothetical example, an employee's performance can directly influence the quality of the product. We can imagine that a good product influences the self esteem of the employee as well which eventually impacts their performance. This can go on in a loop such that we can imagine a steady state between these two observed variables which is represented by a feedback loop in the relational model. We will add this explanation.
>
> > Related work is hard to follow for someone unfamiliar with relational causal models and cyclic models. Consider integrating it with the Preliminaries section.
>
> Thank you for that feedback.
>
> > Definition 1: should be d-separation instead of $\sigma$-separation.
>
> Indeed it should be d-separation instead of $\sigma$-blocked. Thank you for catching this unintentional mistake.
>
> > Why are there more than one relational skeleton?
>
> A relational schema can result in multiple instances. For example, we can assume a simplistic schema with an entity Person and a relationship Friendship which represents friendship ties among instances of entity Person. In one possible instance a person Alice can be friends with only one other person Bob, however in a different instance Alice can be friends with two other persons, Bob and Chris. Both these instances are valid and come from a single schema. Each of these instances can be represented as a separate skeleton.
>
> > What is the difference between [Alice].salary and [Laptop, develops, Alice].salary?
>
> [Alice].salary describes the salary attribute of Employee instance Alice from the perspective of employee class Employee. On the other hand, [Laptop, develops, Alice].salary describes the same instance-attribute pair but from the perspective of entity class Product. While they refer to the same thing in this example, the concept of perspective allows an expressive semantic for referring to relational variables. For example, we can refer to the set of employees who worked on a specific product from the perspective of Product but not from the perspective of Employee. We will include a further discussion about “perspectives” of relational models which should make this clearer and relates to the next point.

---

### Official Review · Reviewer_BnHP · 2021-11-24

**Confidence:** 3
**Overall Score:** 6

**Main Review:**

The focus of the work is on relational causal models, which generalize over previous causal models by capturing interactions between non-iid instances. A causal representation named abstract ground graph (AGG) was previously introduced for reasoning in relational models. But this representation does not allow for cycles. The contribution of this work is to generalize that line of work by extending the framework to allow for cycles. The authors introduce a presentation called sigma-AGG. Also introduce relational sigma-separation for reasoning about relational queries and show that it is sound and complete.

A clear motivation for using relational models is not presented and the advantage of using relational models over, say, chain graphs is not clarified by the authors. This paper is of interest only to readers sufficiently interested in the original papers on relational models.

The authors show that under Assumption 3, relational sigma-separation is complete. The authors have not clarified what is the interpretation of this assumption and what real-life models are excluded due to this assumption.

Definition 3 is not clear. What does it mean that dependencies construct cycles. At this point of the paper, nothing is said yet about the connection between the distribution and a cyclic graph.

In general, the notion of cycle in relational models is not clear. In SCMs, we start with structural equations, and we obtain the edges based on those equations. Then those equation can result in cycles, which could be interpreted as steady state of the equations. What is the meaning of cycles in a relational model?

**Summary:**

---

> ### Author Response · Authors · 2021-12-04
> **Response to reviewer BnHP**
>
> We thank the reviewer for the feedback and provide responses to each point.
>
> > A clear motivation for using relational models is not presented and the advantage of using relational models over, say, chain graphs is not clarified by the authors.
>
> The main motivation for our work is to enable the study of social phenomena through rigorous reasoning and identification of effects of interest (e.g., contagion). Some of the literature on interference and SCM considers only templates with pairwise relationships (E.g., Ogburn & VanderWeele 2014, Shalizi & Thomas 2011). This is unrealistic because social networks are not composed of pairs of nodes that are independent of other pairs of nodes. To be realistic, these templates would need to have dependencies with themselves. RCMs allow us to represent such more complex dependencies. What RCMs offer in this context is two-fold: 1) they offer a general expressive language to reason about causal dependencies in relational data where a single template can represent infinite instances. 2) They allow a rich set of tools (i.e. relational d-separation, AGG) which enable representation and learning of relational causal models from observational samples.
>
> Chain graphs represent feedback as explicitly non-causal edges whereas the framework of $\sigma$-separation also provides a rich representation for reasoning over cyclic relationships. It allows a cycle between two nodes to be represented as an explicit causal relationship, and there can be directed cycles of arbitrary length (chain graphs require a weakened form of acyclicity).
>
> > This paper is of interest only to readers sufficiently interested in the original papers on relational models.
>
> Even though the community where this work directly fits is small, it can be beneficiary to a wide range of audiences. We hope our work can serve a similar purpose to identification in non-relational settings. Researchers from network science and computational social science often want to measure certain influence/peer effect quantities or events and are unsure whether it's possible or not. They can use our work as a tool to identify what can be measured.
>
> > The authors show that under Assumption 3, relational $\sigma$-separation is complete. The authors have not clarified what is the interpretation of this assumption and what real-life models are excluded due to this assumption.
>
> Assumption 3 is on the topology of the *relational* structure, i.e., the network which defines which entities are connected to each other. The interpretation is that entities that are not connected to any other entity or are connected to only one other entity are excluded. For example, if the entities are users in a social network, the framework would only consider users who have degree at least two, i.e., are connected to at least two other users. Assumption 3 prohibits one-to-one or one-to-many relationships and only allow many-to-many relationships which covers a wide range of real-world phenomena. For example, friendship ties in social networks, employers developing products, social network users and their platforms of choice are some common examples which fall within the scope of our work.
>
> > What does it mean that dependencies construct cycles. At this point of the paper, nothing is said yet about the connection between the distribution and a cyclic graph.
>
> We use dependency to denote that the distribution of one random variable is causally influenced by the values of another. The connection between the structure of the graph and the underlying distribution is made via $\sigma$-faithfulness (Def. 3). We will update the text to make this more clear.
>
> > What is the meaning of cycles in a relational model?
>
> The meaning of cycles in relational models is very similar to the meaning of cycles in SCMs except that in relational models, the cycles can be between relational variables and we need to account for the fact that independencies can break when unrolling the relational model at the instance level. This is why relational d-separation was introduced in the first place. A person being influenced (e.g., in their buying habits, political views) by their peers and vice-versa is an example of a feedback loop between relational variables. One way to try to represent these feedback loops would be with SCM templates with pairwise relationships (e.g., the ones used in Ogburn & VanderWeele 2014, Shalizi & Thomas 2011) and assume SUTVA at the node pair level. However, this is not a realistic representation because social networks are not composed of pairs of nodes that are independent of other pairs of nodes. To be realistic, these templates would need to have dependencies with themselves. This is where RCMs help. The fundamental challenge in representation and learning of RCMs is to be able to give guarantees about causal effect identification based on the unfolding of relational templates which can have dependencies with themselves. We will clarify this further.

---

### Official Review · Reviewer_Prcm · 2021-11-24

**Confidence:** 2
**Overall Score:** 6

**Main Review:**

## major comments
1. I wonder how meaningful it is to study causal models with cycles. Can we always decompose the variables s.t. the cycles can be avoided? For example, we can decompose performance into P1 and P2, then we let P1 be the cause of quality and P2 be the effect of quality.
2. It seems assumption 3 is likely to be violated in many cases. For example, in a standard probabilistic SCM, every observed variable would have an exogenous noise variable pointing to it.

## details
1. I think in definition 1 it should be d-separation instead of \sigma-separation.
2. There are many terminologies and notations introduced, it is better to have a table to summarize them.
3. It is better to intuitively introduce the difference between \sigma separation and d-separation with an example after introducing their definitions.



**Summary:**

This work is the first to work on relational causal models. It proposes \sigma-separation for such causal models and show it is sound and complete under relatively strong assumptions.

---

> ### Author Response · Authors · 2021-12-04
> **Response to reviewer Prcm**
>
> We thank the reviewer for the feedback and provide responses to each point.
>
> > I wonder how meaningful it is to study causal models with cycles. Can we always decompose the variables s.t. the cycles can be avoided? For example, we can decompose performance into P1 and P2, then we let P1 be the cause of quality and P2 be the effect of quality.
>
> It might be possible to avoid a cycle by decomposing variables. However, the motivation behind the study of cyclic causal models often comes from a practical point of view where we have limited scope of observation. As a general example, in an economic system, prices of products can be a function of supplied quantities. On the other hand, supplied quantities can also be a function of price. The underlying dynamic process can be described through an acyclic causal graph over time. But if we don’t have enough observations over time and want to approximate such phenomena it eventually introduces a cycle that represents the steady state. Similarly, in our problem setup, we are given a relational schema that consists of the attributes that we can observe. So, even though decomposing variables is an option to avoid cycles theoretically, it cannot necessarily serve the practical purpose in general. We refer the reviewer to [Bongers et. al., 2021](https://arxiv.org/pdf/1611.06221.pdf) for further discussion. We can clarify this further in the paper.
>
> > It seems assumption 3 is likely to be violated in many cases. For example, in a standard probabilistic SCM, every observed variable would have an exogenous noise variable pointing to it.
>
> Assumption 3 is on the topology of the  *relational* structure, i.e., the network which defines which entities are connected to each other. For example, if entities are social network users, this would exclude users of degree 0 or degree 1. However, there is no assumption on the dependence structure over the *random variables*. Under our assumptions it is entirely valid to have each random variable in the ground graph to have an exogenous noise variable pointing to it. We will clarify this further in the paper.
>
> > I think in definition 1 it should be d-separation instead of $\sigma$-separation
>
> Indeed it should be “d-separated” instead of “$\sigma$-blocked” and “d-blocked”. Thank you for catching this unintentional mistake.
>
> > There are many terminologies and notations introduced, it is better to have a table to summarize them.
> > It is better to intuitively introduce the difference between $\sigma$-separation and d-separation with an example after introducing their definitions.
>
> We appreciate the suggestions on improving the paper. We will include a terminology/notation table and an example showing the difference between $\sigma$-separation and d-separation.

---

### Decision · Program_Chairs · 2022-01-12

**Decision:**

Accept (Poster)

**Comment:**

The paper develops an abstract representation to reason about relationships in relations models which may contain cycles. The paper proposes \sigma-separation for such relational causal models. The implication of the proposed work is that there can be directed cycles of arbitrary lengths which is an interesting contribution.

There are certain suggestions/improvements suggested by the reviewers, which could improve the readability of the paper:

(a) an example showing the difference between \sigma-separation and d-separation. (b) more discussion surrounding how the proposed framework is useful as compared to the scenario when the underlying dynamic process can be described through an acyclic causal graph over time (as a cycle can be unfolded in time to yield a directed acyclic graph).

I encourage the authors to improve the readability of the paper by taking into consideration feedback by the reviewers.

I believe the proposed abstract representation to reason about cycles is an important contribution, and would be appreciated by the community.